# Comparison of Four Real-Time Polymerase Chain Reaction Assays for the Detection of SARS-CoV-2 in Respiratory Samples from Tunja, Boyacá, Colombia

**DOI:** 10.3390/tropicalmed7090240

**Published:** 2022-09-10

**Authors:** Lorenzo H. Salamanca-Neita, Óscar Carvajal, Juan Pablo Carvajal, Maribel Forero-Castro, Nidya Alexandra Segura

**Affiliations:** 1Laboratorio Carvajal IPS, SAS, Tunja 150003, Colombia; 2Facultad de Ciencias, Grupo de Investigación en Ciencias Biomédicas, Universidad Pedagógica y Tecnológica de Colombia, Tunja 150003, Colombia

**Keywords:** severe acute respiratory syndrome coronavirus 2 (SARS-CoV2), COVID-19, molecular diagnostics, real-time polymerase chain reaction (RT-qPCR)

## Abstract

Coronavirus disease (COVID-19) is an infectious disease caused by SARS-CoV-2. In Colombia, many commercial methods are now available to perform the RT-qPCR assays, and laboratories must evaluate their diagnostic accuracy to ensure reliable results for patients suspected of being positive for COVID-19. The purpose of this study was to compare four commercial RT-qPCR assays with respect to their ability to detect the SARS-CoV2 virus from nasopharyngeal swab samples referred to Laboratorio Carvajal IPS, SAS in Tunja, Boyacá, Colombia. We utilized 152 respiratory tract samples (Nasopharyngeal Swabs) from patients suspected of having SARS-CoV-2. The diagnostic accuracy of GeneFinder^TM^ COVID-19 Plus RealAmp (In Vitro Diagnostics) (GF-TM), One-Step Real-Time RT-PCR (Vitro Master Diagnostica) (O-S RT-qPCR), and the Berlin modified protocol (BM) were assessed using the gold-standard Berlin protocol (Berlin Charité Probe One-Step RT-qPCR Kit, New England Biolabs) (BR) as a reference. Operational characteristics were estimated in terms of sensitivity, specificity, agreement, and predictive values. Using the gold-standard BR as a reference, the sensitivity/specificity of the diagnostic tests was found to be 100%/92.7% for GF-TM, 92.75%/67.47% for O-S RT-qPCR, and 100%/96.39% for the BM protocol. Using BR as a reference, the sensitivity/specificity for the diagnostic tests were found to be 100%/92.7% for the GF-TM assay, 92.72%/67.47% for the O-S RT-qPCR, and 100%/96.39% for BM. Relative to the BR reference protocol, the GF-TM and BM RT-PCR assays obtained similar results (k = 0.92 and k = 0.96, respectively), whereas the results obtained by O-S-RT-qPCR were only moderately similar. We conclude that the GF-TM and BM protocols offer the best sensitivity and specificity, with similar results in comparison to the gold-standard BR protocol. We recommend evaluating the diagnostic accuracy of the OS-RT-qPCR protocol in future studies with a larger number of samples.

## 1. Introduction

The outbreak of the coronavirus disease in 2019, also known as COVID-19, the causative agent of which is novel severe acute respiratory syndrome coronavirus 2 (SARS-CoV-2), was first detected in China on 31 December 2019 [1]. It was quickly (30 January 2020) declared a pandemic by the World Health Organization (WHO), becoming the first global public health emergency of the 21st century [2]. COVID-19 is becoming increasingly common in the population. Likewise, fear is generated due to the appearance of new variants, which is why vaccine-mediated immunity is needed to avoid the possible appearance of new variants capable of escaping the immune system [3]. Given the high speed of spread and the high cost of health services associated with this viral disease, as well as the lack of effective treatment, diagnosis is essential with continuous evaluation of the kits offered on the market, as well as strategies to generate safe and effective vaccines for vulnerable and susceptible populations and improve response and coverage strategies in health systems [4].

Since the pandemic was declared, the National Institute of Health reports that at least 525,609,637 people have been infected, with 6,277,241 deaths reported globally as of June 2022. In Colombia, 6,099,111 million cases have been reported, with 4214 active cases and 139,833 deaths. In the Department of Boyacá, Colombia, where the study population is located, 125,328 cases were reported during the same period, with a total of 2785 patient deaths (National Institute of Health, 2021, https://www.ins.gov.co/Noticias/Paginas/Coronavirus.aspx accessed on 18 August 2022). The viral genome of SARS-CoV-2 was sequenced on 3 February 2020 by the Shanghai Public Health Clinical Center, Fudan University, Shanghai, China. Thanks to the complete genomic sequencing of this new virus, the development of several vaccines and treatments against this viral disease have been developed. Similarly, the complete sequence allowed for the advancement of various diagnostic protocols for the identification of specific sequences of the viral genome. Specificity was improved using molecular techniques, such as PCR and the implementation of duplex PCR. The latter helped to decrease the time necessary to obtain test results and increase the processing capacity of laboratories worldwide [5].

Given that new variants of SARS-CoV-2 present with specific mutations, PCR assays could fail to detect some viral genes. In general, molecular kits should target conserved sites (e.g., genomic sequences that are the least likely to accumulate mutations over time). In this stage of the SARS-CoV-2 pandemic, an unprecedented number of genomes are available that can readily identify suitable candidates within conserved sites for diagnosis. These molecular assays involve protocols and procedures that allow SARS-CoV-2 to be identified using specific genes, such as E, RdRp, S, and N, among others. Although the virus has undergone multiple mutations, several studies have shown that most of the mutations occur in the region that encodes the spike protein in specific sites of its genome. As a result, the development of tests against other viral proteins has been recommended because the use of S might decrease specificity and cause false-negative test results [6].

To address the diagnosis of this viral disease in Colombia, the head of the National Institute of Health had to face the challenge of implementing diagnostic techniques based on World Health Organization (WHO) recommendations and the national guidelines for the laboratory surveillance of respiratory viruses. Therefore the diagnosis of SARS-CoV-2 is based on the regulations stipulated by the WHO, which recommended the Charité Berlin protocol as the gold standard in diagnostic laboratories, which was implemented in Colombia under the supervision of the Colombian National Institute of Health in collaborating laboratories (https://www.ins.gov.co/Pruebas_Rapidas/2.%20Protocolo%20Est%C3%A1ndar%20para%20validaci%C3%B3n%20de%20PR%20en%20Colombia.pdf accessed on 18 August 2022). Currently, several commercial kits are offered for the molecular diagnosis and identification of SARS-CoV-2 in Colombia, and laboratories must evaluate the diagnostic accuracy of these kits to ensure reliable results with respect to suspected COVID-19 patients.

The objective of this study was to compare four commercial RT-qPCR assays for the detection of the SARS-CoV2 virus using nasopharyngeal swab samples referred to Laboratorio Carvajal IPS, SAS in Tunja, Boyacá, Colombia. This research was approved by the research ethics committee of the Universidad Pedagógica y Tecnológica de Colombia, which was provided in the city of Tunja on 18 November 2021.

## 2. Materials and Methods

### 2.1. Study Design

A single-center prospective study was performed on 152 samples from female (70) and male (82) patients suspected of SARS-CoV2 infection in the department of Boyacá, Colombia. Samples with evidence of inadequate storage; presence of microbial, fungal, or chemical contamination in solvents or reagents; or a volume less than 250 µL, as well as alterations or modifications in labeling, were excluded.

### 2.2. Sample Collection and Preservation

A total of 152 samples from suspected COVID-19 patients were collected from the upper respiratory tract using nasopharyngeal swabs [7]. Samples were immediately placed into sterile tubes containing 3 mL of viral transport medium (VTM) [8]. The samples were stored at a temperature between 2 °C and 8 °C and sent to the molecular biology laboratory of Carvajal Laboratorio IPS SAS to confirm the presence or absence of viral RNA.

We followed the guidelines established by the National Institute of Health of Colombia (INS) (https://www.ins.gov.co/buscador-eventos/Informacindelaboratorio/LineamientosparalavigilanciaporLaboratoriodevirusrespiratorios.pdf accessed on 18 August 2022), the Ministry of Health and Social Protection of Colombia (Minsalud) (https://www.minsalud.gov.co/sites/rid/Lists/BibliotecaDigital/RIDE/VS/ED/VSP/psps03-lineamiento-bioseguridad-red-nal-lab.pdf accessed on 18 August 2022), and the provisional biosecurity guidelines of “Laboratory for the handling and transport of samples associated with the new coronavirus 2019 (2019-nCoV)” (https://www.minsalud.gov.co/sites/rid/Lists/BibliotecaDigital/RIDE/VS/ED/VSP/psps02-lineamientos-gmuestras-pandemia-sars-cov-2-col.pdf accessed on 18 August 2022) for the reception of samples suspected of SARS-CoV-2 positivity.

The samples were received along with a referral form, “INS Basic Data Report Sheet 346”, which included names and surnames of each patient, date of first symptoms of the disease, date of sampling, type of sample (nasopharyngeal aspirate, bronchoalveolar lavage, necropsy, etc.), and any additional epidemiological or clinical data (https://www.ins.gov.co/buscador-eventos/Lineamientos/345_ESI_Irag_2022.pdf accessed on 18 August 2022). All samples met the criteria listed in the Guidelines for Laboratory Surveillance of Respiratory Viruses and the Manual of Sampling for Microbiological Analysis of the Ministry of Health of Bogotá [9].

### 2.3. RNA Extraction

Samples were processed at the same as the kits, so it was not necessary to thaw and freeze them again, thus avoiding RNA damage. RNA was extracted from the virus was with a Nextractor^®^ NX-48S automated nucleic acid extraction system (Genolutium, Gangseo-gu, Seoul, Korea) using an AN NX-48s viral RNA extraction kit (Genolutium, Gangseo-gu, Seoul, Korea) as a reagent, which is compatible with the automation system, following the protocol established by the manufacturer. This system is automated, which favors a reduction in handling errors, prevents cross contamination, and significantly reduces processing times. Viral RNAs were stored at −80 °C for further analysis by RT-qPCR.

### 2.4. Real-Time Polymerase Chain Reaction (RT-qPCR)

In this study, the diagnostic accuracy of GeneFinder^TM^ COVID-19 Plus RealAmp (In Vitro Diagnostics) (GF-TM), One-Step Real-Time RT-PCR (Vitro Master Diagnostica, Granada, España) (O-S RT-qPCR), and the Berlin modified protocol (BM) was assessed. The gold-standard Berlin protocol (Berlin Charité Probe One-Step RT-qPCR Kit, New England Biolabs, Ipswich, Massachusetts) (BR) was used as a reference. Table 1 shows the characteristics of each commercial kit evaluated during the study.

SARS-CoV-2 positivity was confirmed by RT-qPCR using the E, N, and RdRp genes in the GF-TM, O-S RT-qPCR, and BM protocols, respectively. In contrast, the BR reference protocol diagnoses SARS-CoV-2 by the amplification and detection of a region of the E gene, which is shared by various betacoronaviruses of the Sarbecovirus subgenus. In these samples, a positive PCR test was performed in order to detect a specific region of SARS-CoV-2 located in the RdRp gene. For the purpose of reducing these limitations, a duplex PCR test for the detection of the E and RNase P genes, designated the gold standard, was validated. A performance panel was performed with positive and negative samples for SARS-CoV-2 viral RNA. Additionally, the RNase P gene was included as a control to identify the viability of the samples, and to rule out the presence of PCR inhibitors or poor extraction of viral RNA, PCR-grade water was used as negative control for RT-PCR.

For each amplification event, a reaction was performed using a total volume of 25 μL containing 5 μL of RNA extracted in the previous step, 12.5 μL of 2× reaction buffer provided with the Superscript III one-step RT-PCR amplification system with Taq Platinum Polymerase (Invitrogen; containing 0.4 mM of each dNTP and 3.2 mM of magnesium sulfate), 1 μL of reverse transcriptase, 0.4 μL of 50 mM of magnesium sulfate solution (not provided with the kit), 1 μg of non-acetylated bovine serum albumin, and 1.5 μL of each primer from a stock solution of 10 μM. RT-qPCR was performed using a CFX-96 for 10 min at 55 °C, 3 min at 95 °C, 45 cycles of 15 s at 95 °C, and 30 s at 58 °C [10]. The data were analyzed using Bio-Rad CFX Manager software (version 3.1.3090.1022; Applied Biosystems, Waltham, MA, USA). The primers and probe sequences of primers were established by each of the commercial firms based on “Diagnostic detection of 201-nCoV by real-time RT-PCR protocol—Berlin 2020” (https://www.who.int/docs/default-source/coronaviruse/protocol-v2-1.pdf accessed on 18 August 2022).

GF-TM and O-S RT-qPCR were performed according to the manufacturers’ recommendations. The BM protocol was supplied by the Laboratory of Virology at the Universidad del Bosque and modified from the Charité Berlin protocol (https://www.who.int/docs/default-source/coronaviruse/protocol-v2-1.pdf accessed on 18 August 2022) by including a single multiplex PCR reaction for the identification of the E and N genes. All assays used RNA genomic SARS-CoV2, which was provided by the INS or reference laboratories as a positive control (Table 1).

### 2.5. Statistical Analysis

The distribution of the variables was assessed with Kolmogorov–Smirnov test. Categorical data were summarized as absolute frequencies and percentages, whereas categorical variables were summarized as relative and absolute frequencies.

Sensitivity and specificity were calculated in 2 × 2 tables at each level. The sensitivity (95% CI), specificity (95% CI), and positive and negative predictive values were calculated using BR as the gold standard. Matched pairs of recorded cycle threshold values (Ct values) were compared using the Spearman correlation coefficient. Indeterminate results were excluded from the data analysis.

Diagnostic similarities among GF-TM, O-S RT-qPCR, BM, and the gold-standard BR were calculated using accordance analysis with the Fleiss’ Cohen’s kappa (κ) test, in which κ > 0.80 signifies a high similarity between methods. A value of *p* < 0.05 was considered statistically significant. Obtained data were systematized in Microsoft Excel v15.0, and all statistical analyses were performed with IBM^®^ SPSS^®^ 22.0 software (IBM, Armonk, NY, USA).

To evaluate the amplification of genes E and RNase P, the fold change (FC) was calculated using R version 4.2.1 software; the obtained values indicate the number of times that the fluorescence emitted increases or decreases in relation to the reference gene.

## 3. Results

### Comparison of the Results between the Four RT-qPCR Assays

A total of 152 samples from patients suspected of having COVID-19 ranging in age from 1 to 81 years were included in our analysis. A total of 82 samples from men, and 70 samples were from women. Most of the samples belonged were collected from adults between the ages of 25 and 64 years, corresponding to the working age population (57.9%), followed by adolescents between the ages of 15 and 24 years (21.1%), elderly aged 65 years old and older (14.5%), children between the ages of 5 and 14 years (5.3%), and children under 5 years of age (1.3%). Appendix A details the primary data obtained from this study; data were analyzed using OpenEpi ^®^ software.

Table 2 shows the comparative results between the four investigated RT-qPCR assays. Using BR as reference, a total of 152 samples were tested (62 positive and 83 negative), and the sensitivity/specificity of the diagnostic tests was found to be 100%/92.7% for the GF-TM assay, 92.72%/67.47% for the O-S RT-qPCR assay, and 100%/96.39% for the BM assay. Relative the BR reference protocol, the GF-TM and BM RT-PCR assays achieved similar results (k = 0.92 and k = 0.96 respectively), whereas the results obtained with O-S-RT-qPCR were less similar (Table 2). Appendix A details the concordant and discordant results in the 152 analyzed samples.

Figure 1 shows the correlation of Ct cycle threshold values between the RT-qPCR assays for the detection of the SARS-CoV2 virus. We observed a statistically significant strong positive correlation between BM versus BR protocols (r = 0.746, *p* < 0.0001) and between the GF-TM and BR (r = 0.622, *p* < 0.001) protocols. Likewise, there was a significant moderately positive correlation between the O-S RT-qPCR and BR (r = 0.482, *p* < 0.001) protocols.

ROC curve analysis indicated that the best diagnostic kit was the BM, with a predictive capacity of 93%; followed by the GF-TM kit, with a predictive capacity of 87%; and the O-S RT-qPCR kit, with a predictive capacity of 79.7% (Table 3).

In terms of the optimal cycle threshold point evaluated using the Youden index, the BM and GF-TM kits had an excellent specificity and good sensitivity, whereas the O-S RT-qPCR Kit had a good sensitivity but a poor specificity. For the BM Kit, a value greater than 7.2 is considered positive for COVID-19, with a sensitivity of 89.9% and a specificity of 97.6%. Similarly, GF-TM kit values higher than 7.7 are considered positive for COVID-19, with a sensitivity of 79.7% and a specificity of 94%. The O-S RT-qPCR assay is relatively unreliable for the detection of COVID-19, given its optimal cutoff point of 8.4, although it achieves 92.8% sensitivity, whereas its specificity is only 68.7%, meaning that it has a false-positive rate of more than 30%.

To assess the efficiency of the diagnostic kits according to the levels of fluorescence emitted during the amplification process of the E and RNase P genes compared to the gold-standard BR, comparative groups were formed. For group 1, the E gene of BR was taken as a reference; compared to the E gene of GF-TM and the E gene of BM, the efficiency of BR was better than that of BM and GF-TM, as the level of BR fluorescence was 1.4-fold higher than that emitted by the E gene of BM and GF-TM. Similarly, a comparison of the RNase P established that the BR kit fluorescence was 1.3-fold higher than that of GF-TM and BM kits. In group 2, the E gene of the BR was taken as a reference again; however, this time, it was compared with the E gene of GF-TM and the E gene of O-S RT-qPCR, revealing that the efficiency of the E gene of BR was better than that of GF-TM and O-S RT-qPCR, as the BR fluorescence level was 1.3-fold greater than that emitted by the E genes of GF-TM and O-S RT-qPCR. However, for RNase P, the BR kit emitted 0.5-fold more fluorescence than both GF-TM and O-S RT-qPCR. Finally, regarding group 3, the E gene of the BR was evaluated as a reference and compared with the E gene of BM and O-S RT-qPCR; in this case, the behavior of the three kits was similar, as the level of fluorescence of the two compared kits was 0.1-fold higher in relation to that of the BR. In contrast, for RNase P, the BM and O-S RT-qPCR kits presented a 0.7-fold higher level of fluorescence than that of the BR (Table 4).

Table 5 describes the basic advantages and disadvantages of the RT-qPCR assays used to screen for of severe acute respiratory syndrome coronavirus 2 (SARS-CoV-2) with respect to the number of genes detected by the kit, processing time, sample volume, and reagent volume.

## 4. Discussion

Molecular tests based on the identification of specific genes of SARS-CoV-2 that are currently offered on the market and that are used in both symptomatic and asymptomatic patients are characterized by high specificity and low sensitivity, sometimes generating false-negative results [11]. Although the qRT-PCR technique is highly efficient, some studies have shown that it can generate false negatives [11], which could cause a risk to the patient, their family, the community, and the health system, as an infected person could, as a result of an erroneous result, spread the infection. On the other hand, in the clinical setting, it should be clear that the accuracy of diagnostic tests can be influenced by the stage of the patient’s disease and the quality of the samples [12]. In addition, several authors have shown that false negatives can be determined by multiple factors, including poorly trained personnel, incorrectly collected samples, possible errors in the batches of primers and other reagents used for analysis, lack of information from manufacturers, and traceability of the reference materials. Furthermore, it should be emphasized that the RT-PCR technique is not 100% sensitive and specific for other pathogens of importance in the clinical setting, as indicated by data obtained for the diagnosis of SARS-CoV-2 [13,14].

In clinical samples, a positive sample is defined as a Ct value of any specific gene for SARS-CoV-2 of less than or equal to 43; on the contrary, an amplification value greater than 43 is considered a negative result. As an internal control, RNAse P must be present in each sample, and its Ct value must be less than 35 to validate the test. If this gene does not amplify, the test must be invalidated, and the extraction must be repeated (https://www.aidian.eu/uploads/NO-Dokumenter-og-materiell/ES-Products/ELITech/GeneFinder-COVID-19-RealAmp-Plus-Kit_Full-manual_V1_IVD.PDF accessed on 18 August 2022). According to our results, discordant samples were observed in 37 of the 152 samples analyzed. This discrepancy could be associated with the design of the primers by the manufacturers and the fact that the target genes to be identified vary between the diagnostic kits, which might cause changes om the Ct values owing to the amount of RNA assessed; likewise, the viral load of the patient can affect the results. Furthermore, to date, there is no standard methodology, such as calibrators or reference material, among others, that allows for standardization of values between kits offered on the market.

The design of real-time multiplex PCR kits, which identify several target genes in a single reaction, offers lower efficiency in terms of fluorescence levels, in addition to longer amplification times compared to a monoplex PCR, which recognizes a target gene in a single reaction. The most common cause of this effect is the competition for components of the master mix, such as dNTPs and MgCl2, among others, i.e., the more genes of interest identified, as in the case of kits for the diagnosis of SARS-CoV-2 investigated in the present study (BM, GF-TM, and O-S RT-q-PCR, which identify the RdRp, N, E, and RNAse P genes in a single reaction) the lower the levels of fluorescence and the longer the amplification time compared to the kit BR, which identifies only the E and RNAse P genes, which can explain, at least in part, the calculated FC data [15,16].

We evaluated the diagnostic accuracy of four commercially available RT-qPCR methods for the detection of SARS-CoV2 from respiratory samples referred to the Laboratorio Carvajal IPS, SAS in Tunja, Boyacá, Colombia. Our results can help to ensure that tests offered for the screening of SARS-CoV-2 in Colombian patients who are suspected of having COVID-19 meet the criteria for optimal performance.

The current study provides a comprehensive and independent comparison of the analytical performance of primer–probe sets for SARS-CoV-2 testing in several parts of the world. Our findings show a high similarity with respect to the analytical sensitivities for SARS-CoV-2 detection, indicating that the outcomes of different assays are comparable. The primary exception to this is the One-Step Real-Time RT-PCR (Vitro Master Diagnostica, Spain) (O-S RT-qPCR), which had the lowest sensitivity of the investigated kits, consistent with the results of a previous study [17].

This study demonstrates that RT-qPCR significantly improves accuracy and reduces the false-negative rate in the diagnosis of SARS-CoV-2 in pharyngeal swab specimens, which represent a convenient and simple sampling method. Furthermore, qPCR is sensitive and suitable for low-viral-load specimens from patients under isolation and observation who may not be exhibiting clinical symptoms. Finally, RT-qPCR can be used to quantitatively monitor patients to evaluate disease progression [18].

We conclude that the GF-TM and BM protocols offer optimal sensitivity and specificity, as well as results to those of the gold-standard BR protocol, possibly due to the design of the primers. We recommend evaluating the diagnostic accuracy of the OS-RT-qPCR protocol in future studies with a larger number of samples. We recommend that laboratories evaluate the diagnostic accuracy of RT-qPCR assays used for the detection of the SARS-CoV2 virus to ensure reliable results for patients who are suspected of being COVID-19-positive.

## Figures and Tables

**Figure 1 tropicalmed-07-00240-f001:**
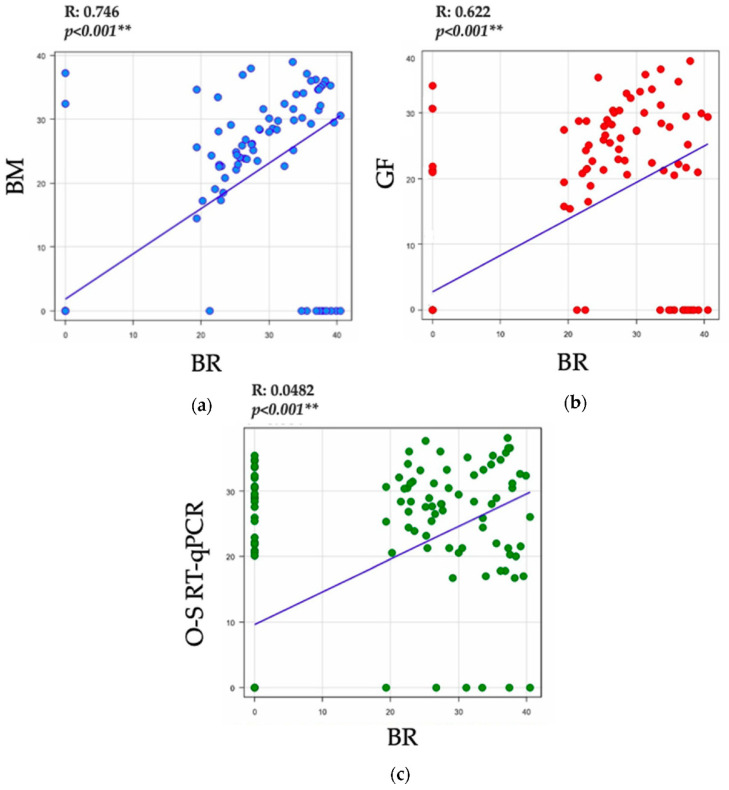
Correlation of Ct values between the RT-qPCR assays for the detection of the SARS-CoV2 virus. (**a**) Correlation of Ct values of E gene between the BM and BR protocols (**b**) Correlation of Ct values of E gene between the GF-TM and BR protocols (**c**) Correlation of Ct values of E gene between the O-S RT-qPCR and BR protocols. Abbreviations: GeneFinder^TM^ COVID-19 Plus RealAmp (In Vitro Diagnostics): GF-TM; One-Step Real-Time RT-PCR (Vitro Master Diagnostica): O-S RT-qPCR; Berlin modified protocol: BM; gold-standard Berlin protocol (Berlin Charité Probe One-Step RT-qPCR Kit, New England Biolabs): BR. (*p*-value < 0.001 ** is highly significant).

**Table 1 tropicalmed-07-00240-t001:** Characteristics of the commercial kits assessed for the detection of SARS-CoV-2.

Feature	GF-TM	O-S RT-qPCR	BM	BR
Manufacturer	In Vitro Diagnostics	Vitro Master Diagnostica	Forest University	New England Biolabs
Sample types	Bronchoalveolar lavage fluid, nasopharyngeal swabs, oropharyngeal swabs, nasal swabs, mid-turbinate nasal swabs, or sputum specimens	Bronchoalveolar lavage fluid and nasopharyngeal swabs	Nasopharyngeal swabs and oropharyngeal swabs	Bronchoalveolar lavage fluid, nasopharyngeal swabs, and oropharyngeal swabs
Sample volume required	5 μL	8 μL	5 μL	5 μL
Extraction required	Yes	Yes	Yes	Yes
Target gene of SARS-CoV-2	E, N, and RdRp	E and N	E and N	And
Internal quality control	RNAse P	RNAse P	RNAse P	RNAse P
Analytical sensitivity	RdRp: 10 copies/test	Gen N: 10 copies/test	Gen N: 10 copies/test	Gen N: 10 copies/test
	N: 10 copies/test	Gen E: 10 copies/test	Gen E: 10 copies/test	Gen E: 10 copies/test
	E: 10 copies/test			
Analytical specificity	1	1	1	1
Maximum performance per kit	100 samples	100 samples	Not specified	100 samples
Test run time	1 h 35′	1 h 2′	1 h 5′	43′
Recommended platform	Biosystems^®^ 7500 Real-Time PCR Instrument (ABI 7500). StepOneTM Real-Time PCR System (Applied Biosystems). CFX96TM Real-Time PCR Detection System (Bio-Rad).	QuantStudioTM 3 Real-Time PCR System (Applied Biosystems). QuantStudioTM 5 Real-Time PCR System (Applied Biosystems). Biosystems^®^ 7500 Real-Time PCR Instrument (ABI 7500). StepOne PlusTM Real-Time PCR System (Applied Biosystems). StepOneTM Real-Time PCR System (Applied Biosystems). CFX96TM Real-Time PCR Detection System (Bio-Rad). Rotor—Gene—Q (Qiagen).	CFX96TM Real-Time PCR Detection System (Bio-Rad).	CFX96TM Real-Time PCR Detection System (Bio-Rad). QuantStudioTM 5 Real-Time PCR System (Applied Biosystems).

Abbreviations: GeneFinder^TM^ COVID-19 Plus RealAmp (In Vitro Diagnostics): GF-TM; One-Step Real-Time RT-PCR (Vitro Master Diagnostica): O-S RT-qPCR; Berlin modified protocol: BM; and gold-standard Berlin protocol (Berlin Charité Probe One-Step RT-qPCR Kit, New England Biolabs): BR.

**Table 2 tropicalmed-07-00240-t002:** Comparison of results between the four molecular assays for the detection of SARS-CoV-2 using the Berlin protocol (BR) as a reference (n = 152, of which 62 were positive and 83 were negative).

Assay		BR Positive	BR Negative	Kappa (k) (±95% cL)	Sensitivity	Specificity	PPV	PNV	GIVES
GF-TM	Positive	69	6	0.92	100%	92.70%	92%	100%	96.05%
	Negative	0	77						
O-S RT-qPCR	Positive	64	27	0.58	92.75%	67.47%	70.33%	91.48%	78.95%
	Negative	5	56						
BM	Positive	69	3	0.96	100%	96.39%	100%	95.87%	98.30%
	Negative	0	80						

Abbreviations: GeneFinder^TM^ COVID-19 Plus RealAmp (In Vitro Diagnostics): GF-TM; One-Step Real-Time RT-PCR (Vitro Master Diagnostica): O-S RT-qPCR; Berlin modified protocol: BM; gold-standard Berlin protocol (Berlin Charité Probe One-Step RT-qPCR Kit, New England Biolabs): BR; predictive positive value: PPV; predictive negative value: PNV; diagnostic accuracy: DA.

**Table 3 tropicalmed-07-00240-t003:** Comparative ROC curve analysis between the RT-qPCR assays for the detection of the SARS-CoV2 virus.

Test Result Variable	Area	Desv. Error ^a^	Asymptotic Significance ^b^	95% Asymptotic Confidence Interval	
				Lower Limit	Upper Limit
BM	93.0%	0.025	<0.001	88%	98%
GF-TM	87.0%	0.033	<0.002	81%	93%
O-S RT-qPCR	79.7%	0.037	<0.003	72%	87%

Abbreviations: GeneFinder^TM^ COVID-19 Plus RealAmp (In Vitro Diagnostics): GF-TM; One-Step Real-Time RT-PCR (Vitro Master Diagnostica): O-S RT-qPCR; Berlin modified protocol: BM; gold-standard Berlin protocol (Berlin Charité Probe One-Step RT-qPCR Kit, New England Biolabs): BR. (Desv. Error ^a^: quantifies the oscillations of the sample mean around the population mean, Asymptotic significance ^b^: degree of compatibility between the proposed population value and the available sample information).

**Table 4 tropicalmed-07-00240-t004:** Fold change of fluorescence levels for genes E and N in three kits assessed in comparison to BR.

Gen/Reference Kit	Gen	Group/Kit	Normalized	Fold Change	Relative Expression	Error
E-BR	E	GF-TM	2.75	0.00	1.00	8.26
		BM	1.30	−1.45	2.73	8.36
	E	GF-TM	2.75	0.00	1.00	8.26
		O-S RT-qPCR	1.43	−1.32	2.49	8.33
	E	BM	1.30	0.00	1.00	8.36
		O-S RT-qPCR	1.43	0.13	0.91	8.33
RNAse P-BR	RNAse P	GF-TM	1.93	0.00	1.00	3.75
		BM	0.61	−1.32	2.50	2.75
	RNAse P	GF-TM	1.93	0.00	1.00	3.75
		O-S RT-qPCR	1.40	−0.53	1.44	6.54
	RNAse P	BM	0.61	0.00	1.00	2.75
		O-S RT-qPCR	1.40	0.79	0.58	6.54

Abbreviations: GeneFinder^TM^ COVID-19 Plus RealAmp (In Vitro Diagnostics): GF-TM; One-Step Real-Time RT-PCR (Vitro Master Diagnostica): O-S RT-qPCR; Berlin modified protocol: BM; gold-standard Berlin protocol (Berlin Charité Probe One-Step RT-qPCR Kit, New England Biolabs): BR.

**Table 5 tropicalmed-07-00240-t005:** Summary of the basic advantages and disadvantages of the real-time polymerase chain reaction assays used to screen for severe acute respiratory syndrome coronavirus 2 (SARS-CoV-2).

RT-qPCR Assay	Advantages	Disadvantages
GF-TM	Identification of three target genes: genes E, N, and RdRp and their reagents	Dependency on commercial company
		Kit for 100 tests
	Easy-to-handle preparation of the reagents	High cost of market availability
O-S RT-qPCR	Short amplification time compared to the other kits: 1 h 2’	Dependence on commercial company
	Identification of two gene targets: E and N	Kit for 100 tests
BM	Identification of two target genes: E and N	Kit for 100 tests
	Easy preparation of reagents for large volumes	High cost of market availability
BR	Reference protocol for molecular detection developed by the Charité Virology Institute	Personnel required to prepare reagents
	Recommended by PAHO for the universal monitoring of SARS-CoV-2 Kit for more than 1000 reactions	Manufacturer outside the country

Abbreviations: GeneFinder^TM^ COVID-19 Plus RealAmp (In Vitro Diagnostics): GF-TM; One-Step Real-Time RT-PCR (Vitro Master Diagnostica): O-S RT-qPCR; Berlin modified protocol: BM; gold-standard Berlin protocol (Berlin Charité Probe One-Step RT-qPCR Kit, New England Biolabs): BR.

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
