# Peer review of "Comparison of Four Real-Time Polymerase Chain Reaction Assays for the Detection of SARS-CoV-2 in Respiratory Samples from Tunja, Boyacá, Colombia"

_tropicalmed, 2022, doi:10.3390/tropicalmed7090240_

Round 1
Reviewer 1 Report
The article "Comparison of Four Real-time Polymerase Chain Reaction As-2 says for the Detection of SARS-CoV-2 on respiratory samples 3 from Tunja, Boyacá - Colombia 4" presents a comparative study which can attract researchers and clinicians concerned about SARS-CoV-2 research and updates. The authors tried to represent a nice piece of work however it must be revised. Suggestions are below.
1. Ethical approval. Authors must disclose the ethical approval number or permission disclosure.
2. Too many short paragraphs, please make them compact, also on if the author could discuss the vaccine and a few drugs in two to three lines will be more attractive to the readers.
3. Line no 100, Can you please give the proper manual link? This link is taken to the ministry website only!
4. Author must perform and present the RQ (Fold change) values and comparative statistical significance or displacement among the samples. Without the RQ values data is not noticeably clear.
5. In Figure 1 why the Y-axis values are inside!
6. Supplementary 1 excel file contains the same column in Spanish and English!!
Author Response
Reviewer 1
Comments and Suggestions for Authors
The article "Comparison of Four Real-time Polymerase Chain Reaction As-2 says for the Detection
of SARS-CoV-2 on respiratory samples 3 from Tunja, Boyacá - Colombia 4" presents a comparative
study which can attract researchers and clinicians concerned about SARS-CoV-2 research and
updates. The authors tried to represent a nice piece of work however it must be revised. Suggestions
are below.
1. Ethical approval. Authors must disclose the ethical approval number or permission
disclosure.
Authors: The information was included as follows “This research was approved by the research
ethics committee of the Universidad Pedagógica y Tecnológica de Colombia, which was given in the
city of Tunja on the 18th of November of 2021.”
2. Too many short paragraphs, please make them compact, also on if the author could discuss
the vaccine and a few drugs in two to three lines will be more attractive to the readers.
The information was included as follows:
Authors: “…COVID 19 disease is becoming more and more common in the population. Likewise,
fear is generated due to the appearance of new variants, which is why vaccine-mediated immunity is
needed to avoid the possible appearance of new variants capable of escaping the immune system
[3]. Given the high speed of spread and the high cost in health services for this viral disease and the
lack of effective treatment, diagnosis is essential with continuous evaluation of the kits offered on the
market, as well as strategies for generate safe and effective vaccines for vulnerable and susceptible
populations and improve response and coverage strategies in health systems [4].”
Line no 100, Can you please give the proper manual link? This link is taken to the ministry website
only!
Authors: All links were verified and the proper links were updated.
3. Author must perform and present the RQ (Fold change) values and comparative statistical
significance or displacement among the samples. Without the RQ values data is not noticeably clear.
Authors: We are currently analyzing data to establish the fold change to compare statistical
significance among samples.
4. In Figure 1 why the Y-axis values are inside!
Authors: The Y-axis legend was fixed in Figure 1
5. Supplementary 1 excel file contains the same column in Spanish and English!!
Authors: The supplementary table was changed
Reviewer 2 Report
The authors compared the performance of 3 real time qRT-PCR diagnostic. Tests for detection of SARS-CoV-2, using the standard Berlin Charité as Gold Standard. Several limitations should be revised before publication. The low amount of new information provided by this study raise the question of its suitability for a full paper or if it would be better included as a Short Communication.
Major concerns
1. The main concern of this study is why several tests are described with non-100% specificity, which raises the question is the Gold Standard was really a Gold Standard?
2. Relatively few samples were analyzed, particularly positive ones (only 69, according to the BR test).
3. Were the samples analyzed simultaneously with the 4 tests? It may be a concern if samples were frozen and thawed different times, since this may affect the possibility of detection of the viral RNA.
4. No information is provided nor discussed on the Ct values of the discordant samples
5. The Discussion is very poor. No discussion on how samples classified as false positive sample sin this study, can be positive by the tests analyzed.
Minor concerns
6. Page 2, line 71-72: It is not correct to refer the molecular tests as against viral proteins¨, please correct, for example, to detecting other viral genes.
7. Very few references are cited and are not updated. In addition, the first two ones does not seem appropriate.
8. Table 1: please correct RNAsaP by RNAse P. Please correct Target region by Target gene and erase Gen in the table, or in any case substitute it by Gene.
9. Table 1: was the analytical sensitivity evaluated in this study? By what method? Of not, please refer as Reported analytical sensitivity.
10. Page 6, Table 1 should be Table 2. Before this Table, it would be useful to include in text how many samples were positive by the Gold Standard BR test.
11. Figure 1 and in other parts of the text: Ct are not cut-off values, but Cycle Threshold Value.
12. Tables 3 and 4 are unnecessary. Some of this information can be included in text.
13. Figure 1: Please increase the text size in the axes.
Author Response
Reviewer 2
Major concerns
1. The main concern of this study is why several tests are described with non-100% specificity,
which raises the question is the Gold Standard was really a Gold Standard?
Authors: The following sentence was included “…Therefore the diagnosis of SARS-Cov2 was based
on the regulations stipulated by the WHO, that recommended the Charité Berlin protocol as the gold
standard in diagnostic laboratories, which was implemented in Colombia and supervised by the
Colombian National Institute of Health to collaborating laboratories
(https://www.ins.gov.co/Pruebas_Rapidas/2.%20Protocolo%20Est%C3%A1ndar%20para%20valid
aci%C3%B3n%20de%20PR%20en%20Colombia.pdf).”
2. Relatively few samples were analyzed, particularly positive ones (only 69, according to the
BR test).
Authors: The RNAs evaluated during the study were analyzed from samples for clinical diagnosis
that arrived at the laboratory, not knowing the absence or presence of genetic material for SARSCoV-
2. The samples were analyzed with the gold standard and then processed with the study kit. Of
the total samples of the day, 69 positive and 83 negative were obtained. All samples were processed
under the same conditions and under the parameters established by each manufacturer in order to
reduce variations in the development of the study.
3. Were the samples analyzed simultaneously with the 4 tests? It may be a concern if samples
were frozen and thawed different times, since this may affect the possibility of detection of the viral
RNA.
Authors: The following sentence was included “Samples were processed at the same time for the
kits, so it was not necessary to thaw and freeze them again, thus avoiding RNA damage…”
4. No information is provided nor discussed on the Ct values of the discordant samples
Authors: The following paragraph was included in discussion “…In clinical samples, a positive
sample is considered when the Ct value of any specific gene for SARS-CoV-2 is less than or equal
to 43, on the contrary, if the amplification is greater than 43, it is considered negative. As an internal
control, RNAse P must be present in each sample and its Ct must be less than 35 to validate the test.
If this gene does not amplify, the test must be invalidated and the extraction repeated
(https://www.aidian.eu/uploads/NO-Dokumenter-og-materiell/ES-Products/ELITech/GeneFinder-
COVID-19-RealAmp-Plus-Kit_Full-manual_V1_IVD.PDF). Based on the results obtained in this
study, discordant samples can be observed in 37 of the 152 samples analyzed. This discrepancy
could be associated with the design of the primers by the manufacturers, and because the target
genes to be identified vary between the kits of diagnosis, which might cause changes of the Ct values
due to the amount of RNA assessed, likewise, the viral load of the patient can affect the results
obtained. In addition to this, to date, there is no standard methodology such as calibrators, or
reference material, among others, that allows standardizing values between the kits offered on the
market…”
5. The Discussion is very poor. No discussion on how samples classified as false positive
sample sin this study, can be positive by the tests analyzed.
Authors: The following paragraph was included in discussion “Molecular tests based on the
identification of specific genes of SARS-CoV-2, which are currently offered on the market, and which
are used in both symptomatic and asymptomatic patients, are characterized by high specificity and
low sensitivity, sometimes generating false negative results [11]. Although the qRT-PCR technique is
highly efficient, some studies have shown that it can generate false negatives [11], which could cause
a risk to the patient, their family, the community, and the health system, since an infected person
could, by having an erroneous result, spread the infection. On the other hand, in the clinical setting,
it should be clear that the accuracy of diagnostic tests can be influenced by the stage of the patient's
disease and the quality of the samples [12]. In addition, several authors have shown that false
negatives can be determined by multiple factors, including poorly trained personnel, poorly taken
samples, possible errors in the batches of primers and other reagents used for analysis, lack of
information from manufacturers, and traceability of the reference materials, in the same way, it should
be emphasized that the RT-PCR technique is not 100% sensitive and specific for other pathogens of
importance in the clinical setting, similar to the data obtained for the diagnosis of SARS-CoV-2
[13][14].”
Minor concerns
6. Page 2, line 71-72: It is not correct to refer the molecular tests as against viral proteins¨, please
correct, for example, to detecting other viral genes.
Authors: The sentence was corrected as follows “These molecular assays involve protocols and
procedures that allow SARS-CoV-2 to be identified using specific genes such as E, RdRp, S, and N,
among others.”
7. Very few references are cited and are not updated. In addition, the first two ones does not
seem appropriate.
Authors: Eleven references were included and reference 1 and 2 were changed
8. Table 1: please correct RNAsaP by RNAse P. Please correct Target region by Target gene and
erase Gen in the table, or in any case substitute it by Gene.
Authors: The table 1 was corrected
9. Table 1: was the analytical sensitivity evaluated in this study? By what method? Of not, please
refer as Reported analytical sensitivity.
Authors: The following information was included “When evaluating the optimal Cycle threshold point
using the Youden Index, the BM and GF-TM kits had an excellent specificity and good sensitivity
while the O-S RT-qPCR Kit had a good sensitivity, but a poor specificity.”
10. Page 6, Table 1 should be Table 2. Before this Table, it would be useful to include in text how
many samples were positive by the Gold Standard BR test.
Authors: The information related to the number of samples was included as follows “Table 1 shows
the comparative results between the four RT-qPCR assays used. Using BR as reference, a total of
152 samples were tested (62 positive and 83 negative)…”
11. Figure 1 and in other parts of the text: Ct are not cut-off values, but Cycle Threshold Value.
Authors: Cut off was change by Ct
12. Tables 3 and 4 are unnecessary. Some of this information can be included in text.
Authors: Table 3 was deleted and table 4 was namely table 3.
13. Figure 1: Please increase the text size in the axes.
Authors: The text size was increased

Reviewer 3 Report
In the submitted manuscript, the authors evaluated 4 different assays as diagnostic tools for SARS COV-2 based on nasopharyngeal swap samples. Although the main objective is interesting, the study is presented in a superficial way and the manuscript is not well-presented. For example, more detains should be included in the methods used in the study "e.g. the primers sequence or reference numbers, enzymes used and the reaction conditions, etc.". Also, there are two “table 1” in the materials and methods and in the results section. Line 27 and 28 is repeated. It is not clear if the authors have used negative controls or not. The discussion needs to be improved based on the authors’ point of view.
Author Response
Reviewer 3
Comments and Suggestions for Authors
In the submitted manuscript, the authors evaluated 4 different assays as diagnostic tools for SARS
COV-2 based on nasopharyngeal swap samples. Although the main objective is interesting, the study
is presented in a superficial way and the manuscript is not well-presented. For example, more detains
should be included in the methods used in the study "e.g. the primers sequence or reference
numbers, enzymes used and the reaction conditions, etc.". Also, there are two “table 1” in the
materials and methods and in the results section. Line 27 and 28 is repeated. It is not clear if the
authors have used negative controls or not. The discussion needs to be improved based on the
authors’ point of view.
Authors:
The use of diagnostic reagents for Sars-CoV-2 in clinical laboratories is based on specific regulations
such as 1036 of 2018
(http://es.presidencia.gov.co/normativa/normativa/DECRETO%201036%20DEL%2021%
20DE%20JUNIO%20DE%202018.pdf) where specific criteria for its intended use are met, many of
the manufacturers, as is the case of the kits used within the protocols, do not reveal the sequence of
their primers, enzymes or other reagents that are used by reagents cataloged as in vitro diagnostics
(IVD).
Methodology and discussion were adjusted
11 references were included
Tables and supplemental material were fixed
We look forward to hearing from you in due time regarding our submission and to respond to any
further questions and comments you may have.

Round 2
Reviewer 1 Report
Dear Authors,
Thank you for your rigorous revision. The revised version did not include the fold change or RQ value comparison in the MS, and not in supplementary also. It should be analysed and compared in the MS.
Author Response
Reviewer 1
Comments and Suggestions for Authors
“Thank you for your rigorous revision. The revised version did not include the fold change or RQ value comparison in the MS, and not in supplementary also. It should be analysed and compared in the MS”.
Authors:
The following paragraph was included in Materials and Methods (lines 203 - 205) “To evaluate the amplification of genes E and RNase P, the fold change (FC) was calculated using the software R version 4.2.1, the values obtained indicate the number of times that the fluorescence emitted increases or decreases in relation to the reference gene.”
The following paragraph was included in Results (lines 273 - 297) “To assess the efficiency of the diagnostic kits by means of the levels of fluorescence emitted during the amplification process of the E and RNase P genes compared to the gold standard BR, comparative groups were carried out. For group 1, the E gene of BR was taken as a reference compared to the E gene of GF-TM and the E gene of BM, it was observed that the efficiency of BR was better than BM and GF-TM, since the level of BR fluorescence was 1.4-fold higher than that emitted by the E gene of BM and GF-TM Similarly, when comparing the RNase P, it was established that the BR kit fluorescence was 1.3-fold higher than that of GF-TM and BM. In group 2, the E gene of the BR was taken as a reference again, but this time it was compared with the E gene of GF-TM and the E gene of O-S RT-qPCR, there it was shown that the efficiency of the E gene of BR was better than that of GF-TM and O-S RT-qPCR, since the BR fluorescence level was 1.3-fold greater than that emitted by the E gene of GF-TM and O-S RT-qPCR. However, for RNase P the BR kit was 0.5-fold higher than both GF-TM and O-S RT-qPCR. Finally, regarding group 3, the E gene of the BR was evaluated as a reference and compared with the E gene of BM and O-S RT-qPCR, in this case, the behavior between the three kits was similar since the level of fluorescence of the 2 kits was 0.1-fold higher in relation to the BR. In contrast, for RNase P, the BM and O-S RT-qPCR kits presented a 0.7-fold higher level of fluorescence than BR (Table 3).”
In addition, table 3 “Fold change of fluorescence levels for genes E and N in three kits assessed when compared to BR.” Was included in results.
The following paragraph was included in discussion (lines 355 - 366) “The design of real-time multiplex PCR kits, which identify several target genes in a single reaction, has a lower efficiency in terms of fluorescence levels, in addition, the amplification times are longer compared to a monoplex PCR, which recognizes in a single reaction a target gene. The most common cause of this effect is the competition for components of the Master mix, such as dNTPs, and MgCl2, among others, that is, the greater the identification of genes of interest, as in the case of kits for the diagnosis of Sars-CoV-2 of the present study BM, GF-TM, and O-S RT-q-PCR, which identify the RdRp, N, E and RNAse P genes in a single reaction, lower the levels of fluorescence and longer the amplification time compared to the kit BR, which identifies only the E and RNAse P genes, which can explain at least in part what is evidenced through the calculated FC data [15] [16].”
Also references 15 and 16 were included as follows:
[15] Kim, H. K., Oh, S. H., Yun, K. A., Sung, H., & Kim, M. N. (2013). Comparison of Anyplex II RV16 with the xTAG respiratory viral panel and Seeplex RV15 for detection of respiratory viruses. Journal of clinical microbiology, 51(4), 1137–1141. https://doi.org/10.1128/JCM.02958-12
[16] Gwyn, S., Abubakar, A., Akinmulero, O., Bergeron, E., Blessing, U. N., Chaitram, J., Coughlin, M. M., Dawurung, A. B., Dickson, F. N., Esiekpe, M., Evbuomwan, E., Greby, S. M., Iriemenam, N. C., Kainulainen, M. H., Naanpoen, T. A., Napoloen, L., Odoh, I., Okoye, M., Olaleye, T., Schuh, A. J., … Martin, D. L. (2022). Performance of SARS-CoV-2 Antigens in a Multiplex Bead Assay for Integrated Serological Surveillance of Neglected Tropical and Other Diseases. The American journal of tropical medicine and hygiene, 107(2), 260–267. Advance online publication. https://doi.org/10.4269/ajtmh.22-0078

Reviewer 2 Report
The authors addressed many of the concerns adequately.
Author Response

(The authors gave the same response as above.)

Reviewer 3 Report
In the resubmitted manuscript, comments have been addressed and the manuscript has been adjusted accordingly.
Author Response

(The authors gave the same response as above.)

Round 3
Reviewer 1 Report
Dear Authors,
Thank you very much for your extensive corrections and for this nice shape of the article.